# Community Bargaining with Urban Growth: The Case of J Village in Guangzhou

**DOI:** 10.3390/ijerph19137857

**Published:** 2022-06-27

**Authors:** Yingyan Xu, Xiaoxing Huang

**Affiliations:** 1School of Public Administration, Guangdong Research Center for Nonprofit Organizations, Guangdong University of Foreign Studies, Guangzhou 510006, China; xuyingyan1102@163.com; 2School of Sociology and Anthropology, Xiamen University, Xiamen 361005, China

**Keywords:** urban growth, community autonomy, vulnerabilities, bargaining with urban growth

## Abstract

The past several decades have seen China undergo a rapid urbanization process. During periods of economic prosperity, cities expropriate outlying lands, often villages, for economic development with the support of various local and national government programs designed to encourage urban expansion. However, the autonomy of the villages has not been paid enough attention. How does incorporation into an urban development zone affect the community identity and autonomy of a village? How does the village bargain with external urban institutions? This research is based on ethnographic research and interviews conducted in 2013, 2014, 2017, and 2021. The results reveal that villagers are generally willing to accept the loss of their collective land in exchange for a larger share of the promised prosperity of industrialization, but over time they tend to find that the immediate benefits of expropriation are outweighed by long-term costs. They lose the support of the state and are exposed to new vulnerabilities, such as pollution and economic instability. Indeed, they agree to undertake unknown future risks in exchange for short-term gains. They cannot gain the right to the city, but gradually lose control of the village.

## 1. Introduction

As reflected in the development policies of a country, urbanization may take different routes to affect and determine the survival or decline of a village. Urbanization involves changes in the boundaries and power relationships between urban and rural areas, as well as the processes of urban governance at different scales. Most theories of urbanization focus on the production of urban space [1], cities as growth machines [2,3], the right to the city [4,5,6], collective consumption, and urban social movements [7,8]. Researchers have proposed the idea of the ‘right to the city’ as a way to respond to neoliberal urbanism and better empower urban dwellers [4,6]. However, in the urbanization process, the villagers may neither gain the right to the city, nor the right to control of their village.

China’s system of regional development aims to promote urbanization through industry and replace the irrational system of a collective economy with rational planning and concrete social space via abstract spatial planning. This approach often has a dual, contradictory effect on urbanized villages: Although the construction of development zones boosts the village economy, the village becomes susceptible to economic vulnerability when it cedes the ownership and autonomy of most of its land in exchange for economic development. In the subsequent development process, the community hopes to regain limited autonomy by negotiating with the external control structures to whom they have ceded ownership. However, in this process, they are entering into a bargain where they agree to undertake unknown future risks in exchange for short-term gains. Rather than depicting the linear process of rural–urban transformation, this paper aims to examine the development of a village at different times and on different scales. The research questions of this paper are the following: How does incorporation into an urban development zone affect the community identity and autonomy of a village? How does the village bargain with external urban institutions? A longitudinal case study is used to analyze these questions: J village, a village undergoing rapid urbanization in the Guangzhou Development Zone, which was surveyed in 2013, 2014, 2017, and 2021.

## 2. Analytical Framework

The transformation of urban space is closely related to urban growth and affects the development of different communities in the city. Molotch proposed the thesis of “the city as a growth machine” to describe the connection between community power and urban structure that had long been overlooked in traditional studies [2]. This theory emphasizes the important role of the government as a dynamic political force for development, which is a kind of “politics of distribution”. The land-based elites make up what Logan and Molotch termed “place entrepreneurs”, defined as “the people directly involved in the exchange of places and collection of rents” [9]. Different types of place entrepreneurs, such as active entrepreneurs and structural speculators, are relevant for understanding the growth machine.

The growth machine theory, which has its roots in spatial and ecological perspectives, posits that allied elites intervene in the land via social organizations to promote urban growth [10]. Nevertheless, as Cain [11] observed, its core claim remains unchanged: “Cities are conceptualized as growth machines, which consist of unified and powerful growth coalitions. These coalitions pursue a pro-growth agenda, seeking to enhance the exchange value of local land and property”.

The concept of urban growth coalitions addresses various urban factors, such as the layout of neighborhood facilities including hospitals and orphanages, whereas the human ecology and growth machine perspectives look into “aspects of facility location” [12], as well as the process of policy making, as elaborated in Gonzalez’s research on the establishment of automobile emission standards in California and the connection between urban growth and the politics of air pollution [13]. Some scholars have also analyzed transformations in the roles of elites in growth coalitions and in their roles in the politics of marketing the city [14]. The growth machine model was designed for rational urban development underpinned by rational planning and regulation of land use, for example, “regulating growth by zoning and building density” [15]. Coalitions of urban elites, a central part of the growth machine, are comprised of economic elites, public officials, scientists, policy specialists, and interest groups [13].

The growth machine theory highlights the driving force of growth coalitions and the leadership of land elites, but it pays little attention to the community. Although the theory also points out that profit-oriented land use may give rise to different communities, such as neighborhood associations that take the government as the arena in which to compete for developmental resources, it places more emphasis on how elites lead the community [2]. However, La Gory suggests that elites are not always allied to enhance growth but may “attempt to discourage urban growth” instead [10]. This results in the emergence of pro-growth and anti-growth interest groups [16]. The success of the growth machine theory does not prescribe the politics that guide development, especially when the costs and benefits of development are hidden from the public. Government officials “sell” many development projects to the communities “on a go/no go basis” with the hope that development will benefit the entire community (such benefits are often distorted by the landowning elites); and, in many cases, development is also accompanied by many negatives, such as higher housing prices, increased pollution [17], and higher taxes [18]. In later urbanization research, the interests of residents and communities have received increasing attention. As the government urges urban growth, community-level agreements and benefits are gradually increasing [11]. Growth machines often encounter opposition from local residents who are more inclined to maintain ownership over the land than sell it [19]. In order to obtain benefits, different communities begin to seek new strategies to address development projects in their backyards. The uneven impacts of urban growth on the community, such as those caused by various local factors and the exclusionary nature of development policies, necessitate examination of community characteristics, as “policy is related to change in census measures of racial, ethnic, and social factors” [20]. However, the existing literature mainly emphasizes the response of different groups in the community to growth and ignores the changes in community autonomy in relation to different attributes and dimensions, especially in different time periods. In some stages, the community actively negotiates with external growth machines, exchanging community development for autonomy, while at other points, the community expects to retain autonomy and lead its own development, but “tends to be unsuccessful in the face of larger-scale commercial development” [11].

The growth machine theory has also been employed, to a certain extent, to analyze the development of different cities in China [21,22,23]. When reflecting on the nature of the city in the context of China’s urbanization, questions of definition, distinction, and integration have become the predominant issues pertaining to land [24]. An understanding of China’s public land ownership regimes is dependent upon an understanding of the difference between “collective” (jiti) and “state” (guoyou) ownership [25]. This distinction remains central to the process of rapid urbanization, as it allows the state to control the reclassification of space and people [26]. This distinction is very simple but relies on a series of important features of China’s spatial organization and management bureaucracy. In short, land is divided into different categories of commodification, such as tradable or non-tradable in different markets and available or unavailable for different functions (e.g., production, construction, and farming). State-owned land can be traded and used for production or construction. However, collectively owned land is strictly restricted to farming or collective production, and it is non-tradable. This distinction is also crucial for explaining the urbanization strategies of regional government initiatives and the resistance of villagers in urbanization areas. Research on major cities as growth machines has yielded fruitful results [15,23]. Studies in this regard stress the vital role of the local government as “local construction entrepreneurs”. Government officials are parallel to city managers, who steer the city machines towards the formation and operation of growth machines.

This paper centers community bargaining in discussion and analysis of urban growth in China. In view of urbanization, the state—often embodied by city governments—seeks to gain control over collective land for both legal and political reasons, such as to promote economic development or to fund its operations by appropriating surplus land value [27]. Unlike emphasizing exchange value with regard to growth machines, the urban government in China takes more factors into account, such as use value, ideology, public sentiment, and so on. On the contrary, villages may apply different strategies to interact with external actors. This makes the urbanization more complicated. In order to explore this issue, this paper focuses on urban–rural fringe areas that are undergoing rapid industrialization and urbanization and combines theories of urban growth, urban spatial transformation, and community autonomy to clarify residents’ opinions about urbanization, collective and state land, and the surrounding environment, so as to explain the unique transformation process of the community. The land use regimes and land-based elites play a central role in the urban growth machine.

## 3. Materials and Methods

J village was chosen for its typicality. It is located beside Pearl River Sea-Going Outlets (Please see Figure 1). In the 1980s, with the demise of the communes and the introduction of the household responsibility system, villager households regained some control over their farming activities. Very soon, however, in the late 1980s, the land of J village became a target for the government’s plan to create Guangzhou’s Economic and Technological Development Zone (“ETDZ” below). Much of the 3000 mu (1 mu ≈ 666.7 m^2^) of land available in the village was expropriated by the government for this purpose. This initial expropriation was followed by others during the late 1980s; as a result, the village currently controls only 200 mu. The next major milestone in the history of the village occurred in 2005, when the Guangzhou Economic and Technological Developmental District (“GETDD” below) was established in Luogang. Territorial hierarchy and regional planning also affected J village. J village was included in Luogang in 2005, and then transferred to Huangpu when the two districts were merged again in 2014. In 2021, there were 2429 registered residents in J village, including about 300 elderly people.

This brief history reveals how J village’s territorial, social, economic, and political boundaries have been dramatically rearranged over the past century and most rapidly in the last three decades of rapid industrialization and urbanization. While little of what has happened recently would have happened without the intervention of the state, many other elements have contributed to the makeup of the village assemblage, the distribution of wealth, and the structure of property rights. This village, like many in China, has had to navigate not only the social and cultural transformations brought on by urbanization, but also the moving boundaries between city and countryside that are defined by the state but constantly negotiated and renegotiated by the village itself.

This research comprised a longitudinal study of J village, a village at the center of China’s urbanization whirlpool that is currently undergoing a transition from a fishing village to an urban area. This study discusses its relationships with the larger social structures and socioeconomic processes over a period of several years. In 2013, we, including the authors and four other researchers, conducted fieldwork in J village and interviewed 31 people including village officials and villagers (see Table 1 below). We selected different residents across different jobs and ages. An outline was prepared for the interviews. The questions included general information (age, gender, family situation, employment situation, official role, housing situation, and family income), the stages of land conversion, the development of the collective economy, etc. All the interviewees were living in Luogang district at the time.

At the same time, we made observations in the village, collected historical text materials (such as documents on land acquisition, demolition, relocation, and resettlement), and recorded different scenes in the village. In June 2014, we visited J village and interviewed the cadres of the Residents’ Committee, including Mr. Guo, the Deputy Director of the Residents’ Committee. In May 2015, May 2016, May 2017, and December 2021, we followed up on the progress of the village and made new observations. Interviews were recorded using digital recorder pens on an informed consent basis. We produced written field notes after the interviews and observations.

## 4. Results

### 4.1. Urban Growth, Economic Development, and Change in Household Registration

Progressive land expropriation from the end of the 1980s ended the villagers’ farming and fishing traditions and shifted their demographic designation from “farmers” to “urban residents”.

The Guangzhou Economic and Technological Development Zone was an early national project sponsored by the State Council, established in December 1984 on only 14,400 mu (9.6 square kilometers) of land. The government needed more land to use for the growth of the ETDZ, and J village provided the perfect opportunity because of its proximity to the original site of the ETDZ. In September 1988, the head office of the ETDZ signed a land expropriation compensation contract with the J villagers’ committee for the expropriation of 1716 mu, more than half the total land.

Among the consequences of this transition was that the village territory was de facto integrated into the zone of exception of the ETDZ; as a result, villagers became entitled to the same tax privileges as ETDZ companies, as well as privileged tariffs for education, public health, medical services, and utility pipe installation. The tax privileges of the ETDZ, for example, allowed the village to import 12 vehicles tax-free.

The process promised enrichment for the villagers, at least temporarily, along with liberation from strenuous farm work and provision of resources that otherwise would have been inaccessible to them as farmers in that period. Most of the interviewees recalled that period as a time of “exhaustion from farming” and reported feeling “relaxed” after the expropriation. The farmers went to work in the factories, or did small business near the village, and earned more than before. At the same time, however, the changes had cut them off from the most important asset they collectively owned—the land—and alienated them from the traditional cultural ties that shaped their identities as farmers and fishermen. While the village economy could be said to have been strengthened, the community itself had been weakened, to the point that it now existed as part of a larger assemblage, the ETDZ, and lived by its rules. Once deprived of its own land, the village assemblage had been de-territorialized and integrated into the ETDZ to serve the purpose of industrialization.

### 4.2. Reshaping the Village Economy in the Context of Urban Development

In the subsequent phase, villagers sought to take advantage of the new situation. In 1991, the Guangzhou government issued the “Decision on expanding the reform and opening up of the Guangzhou Economic and Technical Development Zone”, and the ETDZ entered a new phase of expansion that attracted significant capital. The management committee of the ETDZ was at the time located near J village.

The situation provided opportunities for the two hamlets to develop very rapidly. Transport in and around the village was vastly improved and allowed villagers to travel and take employment opportunities outside the village. Gravel roads, often muddy, became paved concrete streets. The village also underwent a “greening” process (lu hua), and public sanitation improved significantly. The village was integrated into the water network, increasing the availability of safe drinking water to residents. The infrastructural improvements and the more modern appearance of the area were among the changes most deeply imprinted in the memories of the villagers.

As a result of the economic reform, the ETDZ expanded and villagers benefited in several ways. As workers, they could earn higher salaries through employment opportunities to which they had gained privileged access as a result of the agreement with the ETDZ. Others went to work in factories outside of the zone or started their own business. Residents largely saw the period leading up to the expropriation as a time of impoverishment and generally welcomed the new situation.

One of the consequences of expropriation was a fundamental change in the nature of the collective economy. Where farming had been the main economic activity, villagers now found it more profitable to rent out their remaining collective assets, such as collective land and collectively owned buildings. These assets rapidly became the primary income-generator of the collective economy. Villagers also began to monetize their individual properties. Five- or six-story houses soon characterized the village’s changing skyline, often built higher than the newly imposed ETDZ regulations permitted. The regulation of the village real estate now fell under the control of the ETDZ, and so farmers had to come to terms with the fact that they could not simply build houses as they were used to doing, according to the customary allocation of land, but instead had to apply for a proper permit and abide by the rules and restrictions of the ETDZ.

The new houses were central to the new and booming rental economy and added significantly to household incomes, well beyond the profits generated by the collective. With factories now established near the village, migrant workers were rapidly becoming the majority of the population as well as a major source of income. Some villagers could rent up to 20 bedrooms to migrant workers and generate six or seven thousand yuan per month in rent. As one villager told us:

“*I have built two buildings. And I make more than six thousand yuan per month [in rent]. I built them to replace the old ones and now I can lodge migrant workers from all parts of the country. My family is mainly relying on the rent, and I reckon we rank as an average [income] family in the village.*” (2013-18-ZLX-F-41)

The collective economy grew steadily with the extension and success of the ETDZ. Dividends distributed to villagers were generated mostly by the interest on the earmarked compensation fees that were held in the village’s bank account, but were now significantly supplemented by the profits of the booming rental economy. The collective created a new economic cooperative (jingji hezuoshe), which used part of the remaining collective land to build dormitories and other residential buildings that it then leased to factories—mostly labor-intensive textile factories—to accommodate their workers. Until 2000, the yearly payments were around ¥10,000 per family.

In 2002, the collective distributed “fixed” shares (gufen guhua) to villagers. To preserve the value of the collective economy and avoid diluting or redistributing it with every demographic change, as was customary (but illegal after the reform of the 1980s) in socialist collectives, it was decided that no new shares would be distributed after this time, for example to newborns. Shares were distributed to individuals according to age. Those born before 1966 were given eight shares. Those born after 1966 got six. Newborns at the time of distribution would get two shares. Absent villagers, who had abandoned their land before expropriation, received nothing. A later recalculation split one share into 35, bringing the total number of shares in the village to one hundred and thirty thousand. There were two types of shares: “community shares” (shequ gu), which entitled holders to full voting rights, and “social shares” (shehui gu), which were generally distributed to women who married outside of the collective and imparted no right to vote.

In the overall process of urbanization, the collective found that the new regulatory environment imposed by the ETDZ challenged its established practices. As a consequence, the village established collective institutions in an effort to protect the interests of community members and the integrity of its territorial boundaries. Expropriation had weakened the margins of the village and integrated it into the regulatory environment of the development zone, and had also exposed villagers to both the opportunities and the vagaries of the regional, national, and world economies.

### 4.3. Environmental Pollution and Community Vulnerability

The continuing economic growth proved to be a double-edged sword. Heavy industry, coal ports, power stations, steelworks, chemical factories, and fuel storage facilities were taking a heavy toll on the environment. At first, the villagers simply assumed that this was the price to be paid for development, adhering to the government’s “development-first” ideology, but then became dissatisfied.

“*The pollution has existed since the building of the coal port, but we didn’t pay attention to it before. We thought the factories built around the port would block the air from reaching the village. When the wind blows, though, especially from the south, the situation is very bad. Now it is summer and coal dust just blew into our village. Look at these desks! If you forget to close the windows, you can write on the desk’s dust tomorrow morning! I am not exaggerating! […] I heard that even the birds die because of the air pollution. I don’t know whether it is true or not, but the bad smell is certainly there.*”(2014-2-DTJ-LIANG)

Coal in particular was a problem. It was shipped in containers, then transported to the power stations along the transport belt in the west of J village, with intense coal dust pollution following in its wake. Some now questioned the advantages of economic growth.

“*I don’t know what improvements we experienced in the village [after we abandoned farming]. I just know that the air pollution arrived, and the environment is now getting worse. There are no benefits [to becoming an urban citizen]. We lost the land and the pollution came in.*”(2013-21-LIANG1-F-42)

Others felt a sense of inevitability.

“*The air is worse than when we were farming. Now, there are factories all around the village… The enterprises have polluted the environment, and there is nothing we can do.*”(2013-11-F1-F-51)

After years of industrial development, the changes in the built environment were significant. Tall buildings and factories surrounded the community, and the polluted air stagnated in the village, resisting even strong winds. Air and water pollution were blamed for a high number of cases of respiratory cancer in the village. In November 2008, the people’s representative of Guangzhou City inspected the coal wharf next to J village. He reported: “Because of the coal wharf and the factories, J community is facing a heavy pollution of dust and noise. The living of the residents and the community development are seriously influenced. Especially, pulverized coal dust, chimney dust, rust dust and noise are the first enemy of the community environment. It makes the community uninhabitable for the people.” (https://bbs.focus.cn/gz/47467/709c75f38d19f149.html, accessed on 2 June 2022). In 2013, while showing us around the community, Mr. Guo, the Deputy Director of the Residents’ Committee, told us that the government had promised that the factories would be the most advanced in the country and would not cause pollution. This was clearly not the case, and Mr. Guo concluded the story by jokingly saying, “We have become used to this smell.” (16 May 2013 field memo).

Despite this, it was not until 2006 that residents began to complain more than mildly or organize any protests. After all, their economic situation had improved markedly after the inception of the industrial zone, and many thought that the positive externality outweighed the negative. The rent and dividends they received from the collective economy and the advantages of being included in the ETDZ were still seen as sufficient compensation.

Around 2008, however, multiple factors contributed to a deterioration of the economic conditions and a change of heart among villagers, and the relocation of the Luogang district government in 2009 to an area far away from the village took significant business away from the area. In 2008, China’s industry was facing the consequences of the global financial crisis, and many factories were abandoned in the area, setting off a chain reaction. Many of the abandoned factories were those that brought in large numbers of migrants who filled the villagers’ rental properties. The rental economy was therefore also affected, and most leasable bedrooms stood vacant. Without income from rental properties, the collective economy struggled to provide, and profits from dividends declined dramatically.

Under the pressure of the economic crisis and the rapidly changing market conditions of urbanization, villagers began to think that selling their land had not been such a good idea after all. Between 1988 and the mid-2000s, the prices paid by the government to expropriate other land had gone up dozens of times. Villagers in J village learned of neighboring villages that had received over ¥300,000 per mu in compensation, compared with the ¥40,000 they had received, and that had been allowed to retain 10% of their land (J village had retained only 6.6%). Villagers could not help but feel cheated.

The earlier satisfaction for having struck a good deal at the time of compensation was all but gone. Villagers felt that they had supported the development of the Guangzhou economy but had “been forced to eat the bitter fruit of pollution” (2013-9-CM2-F-50). Discontent was spreading in the community, catalyzed by growing pollution. In 2008, more than 200 villagers gathered at the gate of the coal port to block the shipments and demand that the coal port and power stations cease operations. The collective action turned into an open conflict and eventually brought the attention of the Street Party Working Committee (Jiedao danggongwei, street-level government) to the issue of pollution. Mass media reporting of the case also encouraged district government leaders to become involved, and as a result, meetings between the polluters and the residents were arranged in the hopes of negotiating environmental concerns.

Mr. Guo explained that obtaining compensation from the culpable companies was almost impossible without government support, as the village held a relatively weak position in the power structure of the area.

“*The complaints were very serious. We conducted a survey of all the consequences of pollution and presented it to the Party secretary. He received us and organized a meeting with the companies. It was much easier to talk to the companies when the secretary was present. The street office had much better access to the companies than we did. We have no idea of how to deal with them!*”(2013-5-GXB-M-44)

State-owned companies preferred to settle with cash payments. One company paid out ¥10,000 in the first year, and then another ¥30,000. Villagers had to deal with the companies individually. When it became apparent that villagers would receive no significant compensation, they demanded more effective protection from industrial pollution. Following the blockade, the port authority built a large fence around the port in an attempt to reduce the spillover of dust. The fence was a risible solution to the problem of air pollution, however, and villagers were growing increasingly concerned.

Later, the shareholding cooperatives operating in the community established a joint development corporation (kaifa gongsi) to enhance the collective economy and assist in claiming compensation from polluting industries. The main task of the corporation was to collaborate and negotiate with the companies on behalf of the village collective, but it also acted as a subcontractor in the hiring of labor for the companies, thus creating new work opportunities for the villagers as a way to compensate them for the pollution. The corporation was a necessary institutional innovation, as the Residents’ Committee alone lacked the power to deal with or make demands of the companies. In the first year of operation, the development corporation collected over one million yuan in compensation (three million over three years was the agreed amount) and signed a large labor contract with the steelworks to place numerous villagers in jobs at the factory. The Residents’ Committee then redistributed the compensation to the villagers, at around ¥400 for each of the 2500 registered members of the collective. Whatever the corporation gained was put back in the coffers of the collective and used for different welfare provisions. In addition to the urban medical insurance scheme, the Residents’ Committee also established an extra insurance that would compensate the villagers above and beyond the provision of the existing scheme. Over ¥700,000 in medical compensation was distributed as part of the scheme. In 2012, the operating income of the collective was ¥921,396.02, including the rent of factory buildings (¥355,159.12), the rent of the dormitory buildings (¥339,159.12), services for the villagers and the factories (¥566,236.90), and other income (¥16,000). The total income of the collective was ¥2,717,938.45 (a very low amount compared with other villages in Luogang at the time). The total of the compensation fee (called welfare income) was ¥1,668,950.70, which was all used for medical insurance, environment cleaning, death grants, etc. Booming rental economies have become a feature of many villages in the Pearl River Delta. Besides privately held houses, villages still hold a portion of the collective land and earn a significant part of their livelihood from leasing the land to factories or real estate projects. In exchange, their collectives shoulder significant costs to provide services to the factories, and as a consequence are exposed to the risks generated by rapid industrialization, not only in local economies but globally as well. The dangers of depending on a rental economy were demonstrated by the hardships faced by many Dongguan villages after the economic downturn of the global financial crisis [28]. Many of these villages found themselves with neither factories nor workers, and with no other means of sustenance. They also found themselves heavily in debt, as a rental economy had required villages to borrow heavily to sustain growing costs for services and infrastructures [28].

These new vulnerabilities were not unique to J village. In similar phases of development, many villages maintain control of their territory and create institutions capable of redirecting resources created locally to their own members, but have little control of the factors that determine and protect their wealth. By holding onto their assets, villagers lose much of their state protection and are left to suffer the consequences of the rapid and uncontrolled industrialization they had unwittingly permitted. They adhere to the dominant understanding of industrial development as inevitable and are trapped between two roles: that of autonomous rural producers relying on a connection to the land, albeit no longer for farming, and that of urban citizens relying on the declining protection of the state. They thus remain at the margin of the urban, despite having lost any remaining connection with the rural, and become highly vulnerable to the administrative decisions of the state, over which they have no control (for example, the creation of the new district and the relocation of the government facilities). The distinction between city and countryside forces residents of urbanizing rural areas to choose one or the other, but ultimately prevents them from achieving the stability of either.

### 4.4. Failure of Relocation and Reconstruction on the Original Site

Since 2009, the policy of renovating the “three olds” (old towns, old factory buildings, and old villages) has become central to the national urban planning effort and has given rise to large relocation projects. The district government had initially decided against implementing this practice in J village, but as the villagers’ protests intensified, the idea of relocation gained momentum. On 28 December 2012, in a notice posted to the villagers, the Residents’ Committee outlined the task of renovating the “three olds” and proposed a mass relocation of the entire community as the center of their strategy. Appealing to the government, it was thought that using the messaging of the “three olds” campaign would more effectively prepare the village for a full-scale relocation.

At the same time, a notice pertaining to the renewal of houses in J village was posted on the bulletin board and implied that the ETDZ would not approve any new public or private construction in the community until the relocation had been completed. This concerned villagers, who thought they might not be able to renovate their houses—which were in many cases in urgent need of repair or rebuilding—and who were already finding it difficult to rent out local property due to the economic crisis and the growing uncertainty about the future of the area. Occupancy rates declined, especially for shop floors, and villagers were missing the income that leasing had provided. The storekeepers, small business owners, and workers kept on moving out of the village, anticipating the relocation. Many villagers were house owners, with their houses unrented. The main jobs of the villagers were security staff, cleaners, small business owners, and laborers.

Planning for the relocation remained the central concern in the village for the three years between 2012 and 2015. Most of the villagers hoped to escape pollution but found the promised compensation and the location of the new residential community to be unsatisfactory. The first site the government suggested turned out to be a former cemetery, which the villagers refused to occupy. A new plan was proposed in early 2014. When the engineers first came to explain the plan, villagers were generally in favor: 82% agreed to move. However, when the plans were further detailed to the villagers, some noticed that the proposed block was in a mountainous area far from any natural source of water. Despite the dramatic changes in the built environment in J village, its residents still saw the village as defined by its location between river and sea, and many still proudly wore the identity of fishermen. The mountain was not for them. In the end, 71% refused the proposal because of the new location.

“*Relocation to a different place is not a good idea, and some of us didn’t agree. After all, J village is a small fishing village. This is a region between river and sea. If we are relocated, the mountain will take the place of the sea. We could not possibly get used to that lifestyle. […] A fish and a bird have completely different lives!*”(2014-2-DTJ-LIANG)

Furthermore, the residents in the new settlement would be required to pay for typical urban fees, such as property management, gas, and other services, something that had not been necessary in the village. Even worse, they would receive only one apartment for themselves and would not have properties to rent out for additional income.

The major issue of contention, however, was the compensation for the villagers’ houses. The government agreed to compensate for four floors, which had originally been approved. Houses built higher than four floors would be considered illegal, and owners would not be compensated for the extra space. Villagers would receive ¥1200 yuan per square meter, but just the simple cost of construction exceeded that price by some margin, and those who currently had a five- or six-story house did not appreciate that they would be forced to downsize.

The decreasing rental income and the suspension of building approvals were also a focus of the villagers’ complaints. The renovation of old houses was seen as a priority by many families, especially those who still only had a two-story house and wanted to add on to it (2013-14-F3-F-48). One villager petitioned for the authorization to knock down and rebuild his house, and although the application had been submitted well in advance, the proceedings were delayed because of the concomitant Asian Games to be held in Guangzhou.

After one further failed attempt at negotiating, the idea of relocating the village was abandoned, and the Residents’ Committee turned its attention instead to improving infrastructures and living conditions in the village. Mr. Guo explained:

“*If the government decided to relocate our village, we would try our best to persuade the villagers [to agree to the terms of the relocation scheme]. But for now, it has been decided that we will not move, and we want instead to rebuild our community, including the infrastructure.*”(2014-1-DTJ-GUO)

In the meantime, Guangdong province had embarked on a campaign to build “beautiful villages” (meili xiangcun), providing funding to villages undertaking beautification projects. Mr. Guo summarized the current mindset of the villagers at the end of our interview.

“*I am not sure what the future brings, whether we will move, and what will happen if we do. Everything remains uncertain. But we do have opportunities here, right? So, for the moment we will try to improve security and environment, and possibly get some funding through the ‘beautiful villages’ program.*”(2014-1-DTJ-GUO)

Another interviewee Mr. Liang told us that the “beautiful villages” program had indeed improved the environment of the village.

“*After 2016, the Huangpu government confirmed that we would not be relocated. The renovation of urban village was the target. So, the government invested much more in the infrastructure of the village, including the repair of memorial archway, the walls of the kindergarten. Before that time, they were ragged. It’s much better now.*”(2017-Liang)

The Residents’ Committee hoped to gain access to some of this provincial funding to improve the quality of infrastructure, such as by installing new fire prevention systems in the village. They also intended to use the limited space available to develop a community collective economy. Pollution remains a problem, and the Residents’ Committee hopes that now that the villagers are not moving out, the government will relocate the factories instead.

In June 2020, the Huangpu government signed a document entitled “The three-year task of demolition and relocation of old villages of Huangpu District and Development Zone”, and the demolition of J village was part of the plans. The Huangpu government allotted a piece of state-owned land for the demolition and relocation project of J village, in order to prompt the benefit of the project. Since then, reconstruction on the original site has become the main goal of the government and the village. The Director of the Residents’ Committee told us that most of the villagers agreed on this plan and voted for the reconstruction on the original site (2021-HYQ interview).

## 5. Discussion

The process of urbanization involves a series of growth projects and space transformations driven by the belief that national economic development can be accelerated through spatial arrangements. Urbanized villages, as exemplified by J village, are the outcome of national economic development and capital operations as presented through space. In this process, spatial changes in various aspects contribute to changes for communities and ultimately push forward community transition.

In China, the distinctions between urban and rural, and public and collective, allow the state and local collectives to engage one another in this controversial arena, in which actors, including villagers, village officials, industrial capital, and collective economic organizations, debate about development, rationality, and identity, and negotiate services, environmental impacts, and regulations. Stories of land acquisition, industrialization, and attempts at relocation and resettlement testify to the significance of spatial planning as regards the reclassification of the community, the integration of the natural landscape, the impact of road construction, the expansion of houses, and the importance of site selection in relocation and resettlement. The final results of such complex engagements are both highly contingent on local characteristics and highly impactful on local culture and lifestyle. In the process of urbanization, the government focuses on the material reality in service of the ultimate goal of urbanization through planning. Urbanization is, at its core, a one-way process of incorporating land into the city to facilitate its growth, and so benefits to the incorporated land, such as improvement of municipal infrastructure, are often incidental products of reforms whose aims target industrialization rather than quality of life. The village’s collective economy is often held accountable for basic facilities and public services, including road construction and basic public safety guarantees.

In this process, the regional government moves beyond the institutional role of simply guiding transitions and avoiding the externalities incurred from urbanization, and promotes capital accumulation through land expropriation, rationalization, and cooperation, while maintaining its own capacity to interpret the spatial hierarchy of ideologies between urban and rural areas. In terms of the strategy of maintaining control and shaping governance units, the state enters the village through administrative and spatial means without considering the original communities or the strengthened communities in the process of transition. Urbanization conceals various relationships and contradictions related to space, and it also involves actors and communities as subjects. Different forces intertwine to shape the spatial characteristics of urban fringes and reconstruct the original community.

The cost of urbanization is thus largely borne by the local community, a phenomenon known as incomplete urbanization [29]. Under these conditions, the community is also subjected to changes under external pressure. External factors require communities to take on more responsibilities and expose them to new vulnerabilities and invisible economic crises. On the one hand, the industrialization of capital agglomeration has brought about enormous negative externalities in the ecological space where the village resides; on the other hand, in exchange, the village needs to provide services for the factory. This means that it becomes more vulnerable to both environmental crises caused by rapid local industrialization and economic crises associated with new connections to the global economy. Unlike urban communities, villages own collective assets and bear many costs of social governance; urbanized villages do not fully surrender these responsibilities, and so must bear the dual costs of rural collectives and urban industrialization. This industrialization is often seen as an excellent idea in the early stages of land acquisition, and many villagers support industrial development and believe it to be necessary. Eventually, however, they find themselves caught in a paradox of identity: they remain autonomous agricultural producers linked to the land, but are no longer farmers, and they have become urban residents, but are denied full access to urban services and advantages. Therefore, despite being separated from agriculture, they do not fully belong to the city. When these dilemmas appear, community residents come together to maintain the cohesion of the village and jointly resist such impositions, demonstrating the continued importance of community in bargaining with the outside world.

J village is a typical case that has gone through various stages of urbanization, reflecting the interactions of the government, place entrepreneurs, the collective, and the villagers. The land of J village was mostly expropriated by the government in the 1980s, and the area of land left for farmers’ homesteads is small and not valuable for development. The bargaining power of the village is limited based on the land interest. Urban villages are in dire need of improvement [30]. As a comparison, another village in the CBD area of Guangzhou, Liede, has achieved more autonomy and bargained more successfully. Liede is located in the center of the CBD, and the rent for the land is much higher than that of J village. The government could (de-)activate urban village redevelopment [23]. In another paper, the author analyzes another village in Shenzhen, namely, G village. While facing the urbanization process, the villagers persisted in bargaining with the companies and the government, and some collective land was returned to them by the government [31]. While analyzing a village in the urbanization process, we should locate the case in its external context, including policies, administrative structure, etc.

## 6. Conclusions

The urbanization of J village demonstrates the outcomes of urban growth and sprawl, and in a sense supports the idea of cities as growth machines. In the redevelopment process, even if land is transformed from “rural” into “urban”, local government actors maintain some control over the transition, a testament to their role as local construction entrepreneurs.

However, when taking a closer look at the process of community transition, community bargaining constitutes another level of urban growth. In the growth machine theory, the concepts of exchange value and use value are the basic categories. The place entrepreneurs focus on exchange value, while the local residents emphasize use value. The case of J village paints another picture, and we should revise the social typology of place entrepreneurs. The government and the companies took various factors into account and negotiated with the collective. On one hand, the government required the companies to compensate the villagers for the sake of stability. On the other hand, the relocation of the village was decided against due to the low benefits of the relocation project. The broader social developmental context should be considered. From the perspective of the collective and the villagers, not only use value was relevant but also the benefit-sharing right of the development. In a traditional top-down view of urbanization, in which the state incorporates marginal space into the “city”, the incorporated villages immediately lose control over their land. In reality, though, most of the transitions are not planned, but are instead produced through negotiation as the collective employs its remaining mobility and standing to adapt to external spatial changes and the loss of autonomy after land acquisition. This is not to say that no losses occur, however. Once incorporated, the community becomes a part of the city and correspondingly surrenders not only its right to collective land use, but also its cultural and historical connections to the land, as evidenced by the interference of urban building regulations with traditional village home-building practices.

Community autonomy is greatly restricted throughout this process, and so urbanized villages are left with a greater economic dependence on external spaces and factors beyond their control. Community autonomy can be divided into different dimensions embedded in different spatial levels. In these different spaces, the community obtains benefits or rights at different stages through negotiation with the urban growth machine. This process of village transition has given rise to the non-urban and non-rural intermediary community form, distinct from the model of growth and linear development proposed by the urban growth machine theory. The urban growth machine pressures villages to put their communities at risk of greater long-term instability in order to obtain short-term gains, both in terms of material land use and in terms of community identity and autonomy. In future, the social dynamics of urban planning and urban development should be taken into account. A plot of land for collective needs should be returned to the village, which may strengthen the autonomy of the village and the ability to deal with external risks, and then residents may gain “the right to the city” [32]. Meanwhile, the Bureau for Municipal Design should include urban villages in the detailed regulatory planning of the district. At present, the Guangzhou government is renovating the rural industry sections of different villages across the city, which will boost the local economy as well as drawing attention to the development of these villages.

In the past, the growth machine theory mostly emphasized the government and entrepreneurs, and how they formed coalitions for urban growth. This paper introduces the dimension of community autonomy into the theory. Villagers adopt various strategies in response to the market environment and external structures [33]. The collective and the villagers bargain with external actors based on multiple issues, rather than only use value. There are some limitations to this research. Although J village is typical in rapidly urbanizing China, it is particular in terms of its location, time of expropriation, population, and so on. A potential direction for future research would be to look at other villages as case studies and to perform a comparative analysis.

## Figures and Tables

**Figure 1 ijerph-19-07857-f001:**
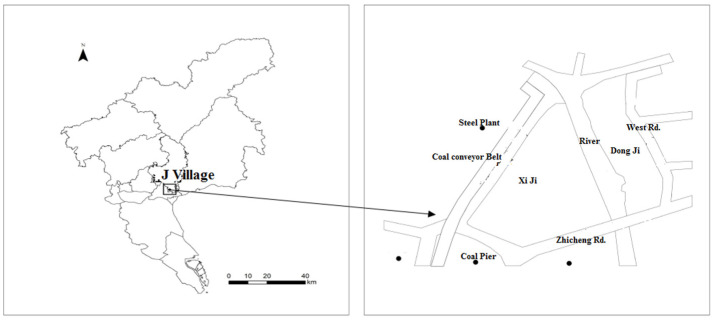
Location and map of J village.

**Table 1 ijerph-19-07857-t001:** Interviewees’ information.

Code Name	Gender	Age	Job	Education	Family Income (Yuan, Yearly)
1-CHEN	F	44	Official of Residents’ committee	College degree	60,000
2-LUO	M	45	Security Guard	Junior high school	
3-MAI	M	38	Social administrator	Junior high school	20,000
4-MAI2	M	51	Security Guard	Junior high school	>10,000
5-GXB	M	44	Deputy director of Residents’ Committee	High school	60,000
6-GS	M	57	Out of work		5000
7-HY	F	56	Retired	Junior high school	20,880
8-CM1	M	45	Security guard	Junior high school	25,000
9-CM2	F	50	Deputy director of third economic team of J village	High school	80,000
10-CM3	M	48	Security guard	Junior high school	55,000
11-F1	F	51	Out of work	Junior high school	10,000
12-M1	M	60	Warehouse keeper	High school	50,000
13-F2	F	42	Out of work	College degree	40,000–50,000
14-F3	F	48	Cleaner	Junior high school	50,000
15-CBL	F	50	Out of work	Junior high school	20,000
16-QYF	F	48	Out of work	Primary school	25,000
17-OU	M	42	Security guard	Junior high school	43,400
18-ZLX	F	41	Deputy director of second economic team of J village	High school	80,000–90,000
19-TANG	F	52	Family planning official	High school	20,000
20-QIN	F	48	Cleaner	Junior high school	18,000
21-LIANG1	F	42	Cleaner	High school	35,000
22-LIANG2	F	48	Worker	Junior high school	72,000
23-HQB	F	39	Cashier	College degree	50,000
24-H	F	57	Housewife	Primary school	50,000
25-GJH	M	44	Worker	High school	20,000
26-XW	M	33	Accountant	High school	100,000
27-QY	F	69	Out of work	Primary school	40,000
28-CMG	M	49	Security guard	Junior high school	40,000
29-GG	F	62	Out of work	Primary school	50,000
30-MY	M	49		Junior high school	60,000
31-CGC	M	50		College degree	80,000

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
