# Peer review of "Community Bargaining with Urban Growth: The Case of J Village in Guangzhou"

_ijerph, 2022, doi:10.3390/ijerph19137857_

Round 1
Reviewer 1 Report
I have reviewed and considered the article entitled "Community bargain with urban growth: The case of J village in Guangzhou". It is a very interesting research, presented in a clear and detailed manner. However, the study only focus on a series of community bargain with urban growth in J village in Guangzhou. Additionally, this research comprised a longitudinal study of J village mainly via the interview. The lack of comparison makes the results unreliable. I highly suggest that authors may conduct some comparative studies and collect vital data to support the argument.
Author Response
Thanks for reviewing. The suggestions are very instructive for the revision of the paper. As a case study, the comparison is lack, which makes the results seem unreliable. We have taken some other cases for the comparison and discussion, such as Liede village, G village.
“These new vulnerabilities were not unique to J village. In similar phases of de-velopment, many villages maintain control of its territory and create institutions ca-pable of redirecting resources created locally to its own members, but have little con-trol of the factors that determine and protect its wealth. By holding onto their assets, villagers lose much of their state protection and are left to suffer the consequences of the rapid and uncontrolled industrialization they had unwittingly permitted. They adhere to the dominant understanding of industrial development as inevitable, and are trapped between two roles: that of autonomous rural producers relying on a con-nection to the land, albeit no longer for farming, and that of urban citizens relying on the declining protection of the state.”
Reviewer 2 Report
This study used the longitudinal materials of Village J in Guangzhou to address the whole process of village bargain with urban growth. The impacts of incorporation into urban development zones on the community identity and autonomy of the village were investigated, through which the bargain process was unfolded. There are several comments:
First, please identify the theoretical contribution of this study, in particular what new could be added on the theoretical of growth machine with this Chinese case?
Second, please address carefully about the research method, how did you choose the interviewees at different time spots, and how did you conduct the interviews.
Author Response
Thanks for reviewing. The suggestions are very instructive for the revision of the paper.
First, we identify the theoretical contribution of this study in the conclusion part. “However, when taking a closer look at the process of community transition, community bargaining constitutes another level of urban growth. In the theory of growth machines, the concepts of exchange value and use value are the basic categories. The place entrepreneurs focus on exchange value, while the local residents emphasize use value. The case of J village draws another picture, and we should revise social typology of place entrepreneurs. The government and the companies took various factors into account, and negotiated with the collective. On one hand, the government required the companies to compensate the villages for the sake of stability. On other hand, relocation of the village was put off for the low benefits of this relocation project. Broader social developmental context should be considered. From the view of the collective and the villages, not only use value was concerned, but also the benefit-sharing right of the development.”
Second, we have added the information of the interviewees, the methods of interviews. Please see table 1.
Kind regards,
Xiaoxing
Reviewer 3 Report
Please provide more information (data) on the dynamics of urban growth of the “J” village during 2013 to 2021. The data you need to provide should include the changes in the compositions of jobs and population. Which jobs are introduced in and moved out from the village? What happened in the village during the study period in terms of the populations age and job profile? You need to add a section for readers to better understand the spatial context. Or, please incorporate these statistics in the Results section. The data should include the changes in job, labor, and property markets, resulted from the external institutional shock, that is the government’s expropriation. I see that you lay out those dynamics throughout the Results section, but please improve those with concrete data.
You need to provide more information on those 31 interviewees: their jobs, demographic information, residential locations, and so force. During the study period, were there significant changes in the status of the people in what aspects?
You concluded that the initial responses to the compulsory urbanization were different from what really happened over time. To support the premise, you should illustrate the evidence by providing longitudinal qualitative observations. What are the Mr Guo’s initial recognition, interim observation, and the recent assessment? What about other interviewees?
You end up with discussing the impact of government-red urbanization policy on the existing rural areas. At the end of the paper, please provide your vision on how the Chinese government should (re)design spatial policies to manage and boost the local economy. And, what are the limitations of the research?
Author Response
Thanks for reviewing. The suggestions are very instructive for the revision of the paper.
First, the jobs in J village did not change much between 2013 and 2021. Many of the villagers kept on renting their apartments and maybe work in the factories, or do small business. We have interviewed the director of residents’ committee in order to follow up on May 30th, 2022. She told us that most of the villagers were; workers in the factories. We add some sentences, “Occupancy rates had declined, especially for shop floors, and villagers were missing the income leasing provided. The storekeepers, small business owners, and the workers kept on moving out of the village, for the relocation expectancy. The villagers were the house owners, with theirs houses unrented. The main jobs of the villages were security staffs, cleans, small business men, and workers.”
Second, we have added the detail information of 31 interviewees. Please see table 1. The jobs of most of the residents did not change, because they had have good education and could not find a better job. Since 2008, the migrant workers had been moving out of the village because of the anticipation of the relocation of the village.
Third, in the interviews with other villagers, they told us that they supported the expropriation of the land at the beginning. And then they built multi-story building for renting. However, the pollution was unbearable. So, they went to protest. Now, they requested for reconstruction on the original place. The responses to the urbanization have been changing over time. We have added some sentences from other interviews. “Another interviewee Mr. Liang told us that the ‘beautiful villages’ program in-deed improved the environment of the village. ‘After 2016, the Huangpu government confirmed that we would not be relocated. The renovation of urban village was the target. So, the government invested much more in the infrastructure of the village, including the repair of memorial archway, the walls of the kindergarten. Before that time, they were ragged. It’s much better now’ (2017-Liang)”
Fourth, the urban renewal project is in progress in Guangzhou. Many urban villages were reconstructed. However, the project focuses mainly on the buildings. Exchange value is emphasized. In the end, the paper mentions the importance of social dynamics. “In future, the social dynamics of urban planning and urban development should be taken into account. A plot of land for collective needs should be returned to the village, which may strengthen the autonomy of the village and the ability to deal with external risks. Meanwhile, the Bureau for Municipal Design bureau should bring the urban villages into the regulatory detailed planning of the district. Recently, the Guangzhou government is renovating the rural industry sections of different villages across the city, which would boost the local economy as well as giving attention to the development of the villages.”
Fifth, in the end, we have discussed the limitation of the paper. “In past, the theory of growth machines almost emphasized the government, the entrepreneurs, and how they got into coalitions for urban growth. This paper brings the dimension of community autonomy into the theory. The collective and the villagers are bargaining with the external actors based on multi discourse, instead of only use value. There are still limitations of the research. Although J village is typical in fast urbanizing China, it is particular for the location, the time of expropriation, the population, and so on. With a comparative view, the researcher did another case studies in different villages, which would be another paper.”
Reviewer 4 Report
Overall good and informative research. It also brings to light transferable qualities that can (should) be applied across global contexts. Perhaps further build this heuristic quality into the conclusions, but otherwise inherent. I do suggest referencing more original sources for some conceptual groundings. For instance, the Harvey work is referring to the original work of Lefebvre's 'right to the city' which could further critically inform the paper and build deeper conceptual (post-Marxist/situationist/structuralist) bearings.
Author Response
Thanks for reviewing. The suggestions are very instructive for the revision of the paper. We have referenced more original sources.
Round 2
Reviewer 1 Report
Thanks for your revising according to previous suggestions. interviewee information in the year of 2013 is also presented. However, several advise below should be considered.
1. Authors can make a detailed and comprehensive comparative study to support your point.
2. Collect vital data to support the argument such as land use or environmental changes.
3. Reference (too old) should be updated.
4. The reliability of the results should be verified by actual material.
Author Response
1. Thanks a lot for the suggestions of the comparative study. We have tried to add more information for the comparing. However, this may not be the main purpose of this paper. We hope to finish another paper for more comparing. 2. The authors have added some vital data of land using and environmental changes. In 2012, the operating income of the collective was ¥921,396.02, including the rent of factory buildings ((¥355,159.12), the rent of the dormitory building (¥339,159.12), the services for the villagers and the factories ((¥566,236.90), and other income (¥16,000). The total of compensation fee (called welfare income) was ¥1,668,950.70, which was all used in medical insurance, environment cleaning, death grant, and etc. . We could find the vial data of the environment changes of J village. However, there were some information about the environmental changes. We have added some sentences: “In November 2008, the people’s representative of Guangzhou City inspected the coal whalf beside J village. He reported as: “Because of the coal wharf and the factories, J community is facing a heavy pollution of dust and noise. The living of the residents and the community development are seriously influenced. Especially, pulverized coal dust, chimney dust, rust dust and noise are the first enemy of the community environment. It makes the community uninhabitable for the people.” 3. We have updated more references. 4. We have added more data and actual material in the paper.Reviewer 3 Report
In the earlier review, I wanted the authors to include some concrete data on the village's socio-economic status. Is the village adminstratively too small for the data to be published in a formal regional database?
I have no further issue on the revised paper.
Author Response
Thanks a lot. We have added some concrete data. In 2012, the operating income of the collective was ¥921,396.02, including the rent of factory buildings ((¥355,159.12), the rent of the dormitory building (¥339,159.12), the services for the villagers and the factories ((¥566,236.90), and other income (¥16,000). The total of compensation fee (called welfare income) was ¥1,668,950.70, which was all used in medical insurance, environment cleaning, death grant, and etc. .